# Transient Polyhydramnios during Pregnancy Complicated with Gestational Diabetes Mellitus: Case Report and Systematic Review

**DOI:** 10.3390/diagnostics12061340

**Published:** 2022-05-28

**Authors:** Agnesa Preda, Adela Gabriela Ștefan, Silviu Daniel Preda, Alexandru Cristian Comănescu, Mircea-Cătălin Forțofoiu, Mihaela Ionela Vladu, Maria Forțofoiu, Maria Moța

**Affiliations:** 1Doctoral School, Faculty of Medicine, University of Medicine and Pharmacy of Craiova, 200349 Craiova, Romania; agnesapcela@yahoo.com (A.P.); mmota53@yahoo.com (M.M.); 2Clinical County Emergency Hospital of Craiova, 200642 Craiova, Romania; sdpreda@gmail.com (S.D.P.); alexcom8000@yahoo.com (A.C.C.); mihmitzu@yahoo.com (M.I.V.); 3Department of Diabetes, Nutrition and Metabolic Diseases, Calafat Municipal Hospital, 205200 Calafat, Romania; 4Department of Surgery, University of Medicine and Pharmacy of Craiova, 200349 Craiova, Romania; 5Department of Obstetrics and Ginecology, Faculty of Medicine, University of Medicine and Pharmacy of Craiova, 200349 Craiova, Romania; 6Department of Medical Semiology, Faculty of Medicine, University of Medicine and Pharmacy of Craiova, 200349 Craiova, Romania; 7“Philanthropy” Clinical Municipal Hospital of Craiova, 200143 Craiova, Romania; maria.fortofoiu@umfcv.ro; 8Department of Diabetes, Nutrition and Metabolic Diseases, Faculty of Medicine, University of Medicine and Pharmacy of Craiova, 200349 Craiova, Romania; 9Department of Emergency Medicine, Faculty of Medicine, University of Medicine and Pharmacy of Craiova, 200349 Craiova, Romania

**Keywords:** polyhydramnios, gestational diabetes mellitus, high risk pregnancy

## Abstract

Polyhydramnios is an obstetrical condition defined as a pathological increase in the amniotic fluid and is associated with a high risk of maternal-fetal complications. Common causes of polyhydramnios include fetal anatomical and genetic abnormalities, gestational diabetes mellitus, and fetal viral infections. We present the case of a 30-year-old Caucasian woman with transient polyhydramnios associated with gestational diabetes mellitus and obstetric complications. The diagnosis was based on the ultrasound assessment of amniotic fluid volume during a common examination at 26 weeks. Two weeks prior, the patient had been diagnosed with gestational diabetes mellitus. After 4 days, the patient was examined, and the amniotic fluid index returned to normal values. At 38 weeks, the patient presented to the emergency room due to lack of fetal active movement. Ultrasound revealed polyhydramnios, the patient was admitted for severe fetal bradycardia, and fetal extraction through emergency cesarian section was performed. Six weeks after birth, the patient underwent an oral glucose tolerance test with normal values, confirming gestational diabetes mellitus. We performed a systematic review of the literature on polyhydramnios, from January 2016 to April 2022, to analyze all recent published cases and identify the most common etiological causes and important aspects related to maternal-fetal outcomes.

## 1. Introduction

Polyhydramnios is an obstetrical condition occurring in 0.2–2% of pregnancies [1,2,3]. Polyhydramnios pregnancies have a high risk of maternal and fetal complications, with a higher incidence of in utero fetal death, premature birth, and the need for emergency fetal extraction [4]. Identifying the etiology of polyhydramnios may play a crucial role in subsequent prenatal medical care and pregnancy evolution [5]. 

Polyhydramnios is defined as a high volume of amniotic fluid for the patient’s gestational age (>95 percentile). Excess amniotic fluid cannot be clinically examined, and the diagnostic criteria are established using ultrasound. Various techniques have been proposed to estimate the quantity of amniotic fluid, the most common being the measurement of either the amniotic fluid index (AFI) or the deepest vertical amniotic fluid pocket (DVP) [6]. 

The AFI method was first introduced by Phelan et al. in 1987 [7], and it entails the division of the maternal womb into four quadrants, having as checkpoints the umbilical scar and the median line. Ultrasound examination is performed by measuring the deepest pocket of amniotic liquid from every quadrant, where a loop free of the umbilical cord is observed, and the sum of these four measurements represents the AFI. Polyhydramnios is defined as an AFI > 24 cm [1].

The DVP method involves the identification and measurement of the largest vertical diameter within an amniotic fluid pocket that is at least 1 cm wide, with excess being defined as values over 8 cm [1].

Approximately 50% of polyhydramnios cases are idiopathic [1]. Etiological causes include [8,9]: gestational diabetes mellitus (GDM), anatomic abnormalities (defects of the central nervous system, gastrointestinal malformations, muscular and skeletal pathologies, urogenital tumors), genetic diseases (trisomy 13, 18, 21), rhesus isoimmunization, infectious diseases during pregnancy (toxoplasmosis, cytomegalovirus, parvovirus), Dandy-Walker syndrome, and Bartter syndrome. 

GDM is one of the etiological causes of polyhydramnios with a variable prevalence [2,9,10]. Various theories have been proposed to explain the onset of polyhydramnios in pregnant women with GDM, the most probable explanation being fetal polyuria caused by hyperglycemia. 

Transient polyhydramnios is a pathology that raises problems in the diagnosis, monitoring, and management of pregnancy [11].

## 2. Case Report

A 30-year-old secundigravida, nullipara Caucasian woman was monitored at the Obstetrics-Gynecology Ambulatory from the 1st trimester. Pregnancy was achieved through in vitro fertilization; at 24 weeks gestational age, due to a premature birth risk, a vaginal pessary was fitted. 

An oral glucose tolerance test (OGTT) was performed at 24 weeks, with the following results: fasting plasma glucose, 70 mg/dL; after 1 h, 180 mg/dL, and after 2 h, 155 mg/dL. A diagnosis of GDM was established. The pathology was treated with diet alone, with no insulin requirement. The patient’s HbA1c level was 5.41% in the 3rd trimester.

The patient was included in a therapeutic educational program in order to acquire the necessary knowledge for self-monitoring of blood glucose (SMBG) for fasting, preprandial, and 1 h and 2 h postprandial values and logging of values in a table and knowledge regarding nutritional medical therapy and decision-making related to alimentary habits. The goal was to achieve the following glycemic targets: preprandial values below 95 mg/dL, 1 h postprandial levels under 140 mg/dL, 2 h postprandial levels under 120 mg/dL. The patient was counseled regarding the negative role of high-glycemic refined carbohydrate ingestion, which may induce postprandial hyperglycemia, and regarding the benefit of nutrient-dense foods with complex carbohydrates in preventing postprandial hyperglycemia

As it was considered a high-risk pregnancy, the patient was examined in the ambulatory service twice a month. At the next appointment, the patient was evaluated by the diabetologist. The patient was informed that high glycemic variability existed due to low adherence to the diet: irregular meals, periods of loss of appetite and nausea followed by periods of anxiety, excessive hunger, and refuge in the consumption of fast-absorbing carbohydrate foods (chocolate, bananas, high-sugar snacks, sugary soft drinks, popcorn, etc.). Glycated hemoglobin (HbA1c) was normal, but HbA1c in pregnancy does not accurately reflect the average blood glucose for 3 months due to increased red blood cell turnover. On the other hand, alternating low blood glucose levels with glycemic target values can lead to normal HbA1c. This makes monitoring blood glucose even more crucial. The patient was further counseled for strict adherence to a balanced diet, divided into 3 main meals a day, to gain proper control. On the same day, an obstetric ultrasound examination was performed, and an excess of amniotic fluid was found, AFI = 27.42 cm (Figure 1).

The patient visited after 4 days for reexamination of the fetal state. The AFI decreased to 18.85 cm, with normal values. Regarding SMBG, a favorable change was seen, with glycemic targets achieved. Periodic examination of the amniotic fluid was performed by the attending physician, and an oscillating evolution of the AFI values was observed until the gestational age of 30 weeks (Figure 2). These oscillations were directly correlated with the maternal glycemic control of the respective period, with the patient invoking various reasons to justify low adherence to the diet. The decision not to introduce drug therapy was made because blood glucose in more than 80% was within normal limits under an adequate diet; each time, however, the patient received the advice on proper nutrition and the need to reach glycemic targets.

Differential diagnosis entailed excluding other potential etiological causes for polyhydramnios: -Blood type and Rh were determined at the first prenatal visit—patient with negative Rh. During pregnancy, the anti-Rh antibodies were undetectable.-TORCH profile (performed during the 1st trimester)—normal.-The patient underwent screening evaluation for fetal abnormalities at the gestational age of 12 weeks and 5 days, with free β-HCG = 1.177 MoM, PAPP-A = 2.050 MoM, and the following risks for trisomy: trisomy 21 (Down syndrome), 1:3987; trisomy 18, 1:9641; and trisomy 13, 1:1514. These values include patients in the low-risk group for fetal genetic abnormalities.-Morphological examination during the 2nd trimester highlights the normal anatomy of the fetus (Figure 3).

-Muscular and skeletal diseases were excluded based on the presence of long bones with a normal shape and echogenicity without any visible asymmetry.-A digestive system with a normal aspect: complete anterior abdominal wall, stomach in a left position under the diaphragm, normal fetal intestinal development with normal echogenicity.-Central nervous system with no symmetric changes and a normal aspect. Complete fetal spine. No neural-tube defects were observed.

During the 32 weeks examination, the AFI was within normal range. During the following examinations, no further noteworthy changes in amniotic fluid volume were observed. 

Diabetes education was paramount regarding the control of maternal GDM; after 32 weeks of gestation, the blood glucose levels were satisfying.

At 38 weeks, the patient presented to the emergency room due to lack of fetal active movement. Ultrasound examination revealed an AFI of 26 cm, fetal cardiac activity, and fetal heartbeat of 90 bpm, highlighting severe fetal bradycardia. No active fetal or respiratory movements were observed during the examination. Considering the maternal GDM, a blood glucose was collected in the Emergency Service, with a value of 260 mg/dL.

The patient recounted that prior to the Emergency Service presentation she had gone on a 7 h trip during which she had been unable to adhere to the dietary plan. The last self-determined glucose value was 60 mg/dL. She had ingested products with a high content of refined carbohydrates, due to hypoglycemic symptoms, without being able to specify the exact amount. 

The patient was admitted to the obstetrics-gynecology clinic for fetal state monitoring and correction of bradycardia.

For acute fetal distress, fetal extraction through emergency cesarean section was performed, delivering a single, live fetus (weight 3460 g, Apgar score 1 min-9, after 5 min-9. During surgery, a specimen of amniotic fluid was collected for microbiological examination, with a negative result. No macroscopic changes were observed in the placenta. Histopathological examination of the placenta revealed a normal placental structure. 

Postpartum, both mother and neonate presented a favorable evolution. The newborn did not present any respiratory distress or other neonatal complications.

Six weeks after birth, the patient underwent OGTT with the following results: fasting glycemia, 73 mg/dL; after 2 h, 106 mg/dL; thus, the diagnosis of gestational diabetes was confirmed, and the results were normal.

## 3. Discussion

The literature describes multiple cases of transient polyhydramnios identified during the 2nd and 3rd trimesters [11]. Variations in amniotic fluid may cause rapid deterioration of fetal well-being, which may lead to intrauterine fetal death. One explanation for the deterioration of fetal health status is placental hemodynamic changes caused by the onset of signs of fetal distress and uncorrectable bradycardia [11,12,13]. 

Our case highlights the monitoring of an early onset of a transient polyhydramnios. It provides information about glycemic control, variations in AFI due to poor control, the importance of self-monitoring, and the maternal-fetal outcome. These findings could help other physicians when encountering this pathology with the same evolution of AFI. The most important thing for achieving good management in pregnancies complicated with transient polyhydramnios is a well-informed multidisciplinary team. In our case, there was good collaboration between the diabetologist and the obstetrician about the need for frequent consultations, assistance with the treatment plan, and counseling of the patient on the benefits of a balanced diet and risks of abnormal glycemic values.

We performed a systematic review of the literature on polyhydramnios to analyze all recent published cases and identify the most common etiological causes and important aspects related to maternal-fetal outcomes. 

To perform this systematic review, we used the Preferred Reporting Items for Systematic Reviews and Meta-Analyses (PRISMA) guidelines. PubMed and Science Direct bibliographical databases were searched for eligible articles using the keyword “polyhydramnios”. Only case reports and case series were considered. Title and abstract screening were conducted independently by two investigators. A third reviewer acted as an arbiter in instances where incongruency of opinion occurred.

Eligibility criteria for inclusion in the study were: all full-text, English-language articles reporting polyhydramnios on singleton pregnancies. The following articles were excluded from this systematic review: articles not written in English, studies on twin pregnancies, or those with no obstetrical information, autopsy specimens. 

This research identified 394 articles published from January 2016 to April 2022.

After removal of duplicates and screening of titles and abstracts, 93 studies were further assessed for eligibility. Ultimately, 35 articles with a total sample size of 47 pregnant women with polyhydramnios were included in this review. The included studies were published between 2016 and 2022. The trial flow diagram is shown in Figure 4.

Data related to age, parity, gestational age at discovery of polyhydramnios, etiology of polyhydramnios, delivery, and maternal and neonatal outcomes were extracted. Additionally, we collected data regarding amniotic fluid index (AFI) or deepest vertical pocket (DVP), gestational age at delivery, birth weight of the neonate, and Apgar score (Table 1).

In the table presented above, we were unable to collect full data according to our primary objective because in some cases, the information was not available.

Twenty-nine studies were case reports, and six were case series. The studies were conducted in South Korea (3), France (1), Germany (1), Portugal (3), China (4), Japan (8), Romania (1), Switzerland (1), the United States (3), the United Kingdom (2), India (3), Spain (2), Taiwan (1), Thailand (1), and Italy (1).

Amniotic fluid was assessed using AFI or DVP. In only two studies, amniotic fluid volume was assessed using the DVP method [14,15]. Six studies mentioned the occurrence of polyhydramnios without providing exact data regarding the measurements. The patients’ characteristics are summarized in Table 2.

The analysis of the evolution of polyhydramnios among the reviewed cases shows a persistent increase in amniotic fluid volume. The mean value for AFI is 36.47 ± 11.02 cm, a much higher value than that obtained during ultrasound examinations performed on our patient. The peculiarity of this case stems from the transient and undulant evolution of the amniotic fluid volume. 

According to the results, most deliveries were premature, with a mean birth weight above the 90th percentile. The delivery of our patient was performed at term, 38 weeks, with an offspring with an adequate birth weight for gestational age. The majority (64.28%) of pregnant women required an emergency caesarean section for fetal interest, and the results are also applicable to our case report: Isolated transient polyhydramnios translated into a maternal unfavorable outcome, represented by the necessity of an emergency cesarean delivery that resulted from poor fetal heart rate tracings. 

The mean Apgar scores at 1 and 5 min were low (5.96 ± 2.53; 7.5 ± 2.09), indicating poor prognosis and unfavorable neonatal outcomes. In fact, only 3 neonates (6.38%) had good clinical condition at birth [16,17,48]. In our case, the neonate had a favorable outcome; the Apgar score was higher than the mean value identified in other studies. 

In the literature, 15 (31.91%) neonates presented an unfavorable evolution with death occurring at variable times from delivery: 3 perinatal deaths (6.38%), 5 neonatal deaths (10.64%), and 7 infant deaths (14.89%). In nine cases (19.15%), death occurred due to respiratory failure; in one case, the parents decided to discontinue the life-sustaining measurements due to very poor prognosis [18], and in one case (2.13%), the neonate was stillborn [19].

The pathologic conditions associated with polyhydramnios are summarized in Table 3. 

After analysis of the case reports of fetuses with associated genetic disorders, we observe that only two out of nine studies reported offspring who were diagnosed antenatally. In other studies, newborns underwent genetic testing and diagnosis postnatally.

Data from this review highlight a low percentage of studies based on polyhydramnios associated with GDM. Baro et al. [44] reported a case of a pregnancy complicated with GDM but also with an associated fetal abnormality, sacrococcygeal teratoma. Comparing this case report with ours, there are some similarities, namely, the good neonatal outcome and the necessity of cesarean delivery due to abnormal fetal heart rate. In our case, the pregnancy was associated with GDM, which appears to be the etiology of polyhydramnios. Maternal hyperglycemia leads to fetal polyuria, with secondary buildup of excessive amniotic fluid. Nevertheless, fetal polyuria can also be associated with Bartter syndrome and other renal fetal anomalies. The presented case, with full anatomy scan during pregnancy and low risk for aneuploidies, emphasizes the importance of the differential diagnosis regarding the etiology of polyhydramnios. Moreover, glycemic control, which is a favorable indicator of maternal-fetal outcomes, can be irrelevant when other fetal pathologies associated are in play. 

Considering the polymorphism of pathologies that can be associated with polyhydramnios, the neonatal outcome can also be variable. 

Neonates confirmed to have genetic disorders had different grades of facial dysmorphism [20,21], with the presence of other anomalies such as: micrognathia, short neck, small dysplastic, low-set ears, and flattened nasal bridge [6]. Furthermore, genetic syndromes with specific common anomalies were identified: in a study about Costello syndrome, all the neonates presented atrial tachycardia, feeding problems, growth retardation, cardiac structural anomalies, and respiratory distress [22].

The association with renal tubular disorder was expressed as serum electrolyte imbalance and polyuria but with a good prognosis after hydroelectrolytic balancing [23,24].

The clinical significance of polyhydramnios is important given its association with congenital fetal abnormalities, unfavorable pregnancy outcomes, and perinatal mortality. 

Given this risk, patient counseling is of great importance, and screening is warranted for congenital abnormalities. Despite screening for fetal anomalies, genetic disorders, and control of maternal diabetes, the increased risk of perinatal mortality persists [49]. 

The presented case represents a challenge for the medical team monitoring the pregnant woman, both for the obstetrician who performs antepartum fetal surveillance by serial sonographic examinations, and for the diabetologist. HbA1c is useful in monitoring maternal metabolic balance, but it does not appear to be the most accurate marker in the second and third trimesters of pregnancy, due to a possible discrepancy between HbA1c values and transient maternal hyperglycemia. Self-monitored blood glucose appears to be a more accurate indicator.

Strict glycemic control is of major importance due to the increased risk of unstable GDM, both maternal and fetal.

Good glycemic control was not, in this case, a sufficient factor for a positive prognosis, as the pregnancy was complicated by the necessity of performing an emergency cesarean section. It may be that isolated transient polyhydramnios is a risk factor especially for obstetrical interventions.

The sub-stratification of pregnant women with GDM based on oxidative status may increase the quality of medical care services and identify patients at high risk of developing complications [50]. 

In modern times, machine learning (ML) may be a complex complementary method of diagnosing and monitoring GDM and preventing negative outcomes [51]. ML is transforming health care systems and may improve treatment plans regarding GDM. There is vast potential in the application of ML to considerably increase the quality of obstetric care and make it more well organized and structured. This case report and the review highlight the scarcity of available data to make adequate judgments, and it emphasizes the need for a registry. The data from the registry may be used in the application of ML in order to identify pregnant women at risk of developing GDM or pregnant women who have undiagnosed type 2 diabetes [52]. This is a revolutionary method that may be used for predicting glycemic events and diagnosing the potential complications. As our case demonstrates, this could help avoid such complications as fetal bradycardia and the necessity of emergency C-section.

## 4. Conclusions

Transient polyhydramnios has an important impact on prenatal care programs. There is no consensus regarding the management of pregnancies associated with transient polyhydramnios. An oscillating evolution of the amniotic fluid quantity and the possible maternal and fetal complications that may occur require personalized approaches to prenatal visits and pregnancy monitoring. A very important role is that of the correct counselling of the patient, who needs to be capable of recognizing alarming signs on time such as lack of perception of fetal movements, as these may be precursors of subsequent severe complications.

It is necessary to eliminate the secondary causes of polyhydramnios, and no less important is the establishment of a multidisciplinary medical team in case of an associated pathology such as GDM.

Systematic monitoring of amniotic fluid volume, identification of high-risk pregnancies, and more thorough monitoring, with more frequent gynecological examinations, may lead to a decrease in the incidence of maternal and fetal complications associated with this pathology.

## Figures and Tables

**Figure 1 diagnostics-12-01340-f001:**
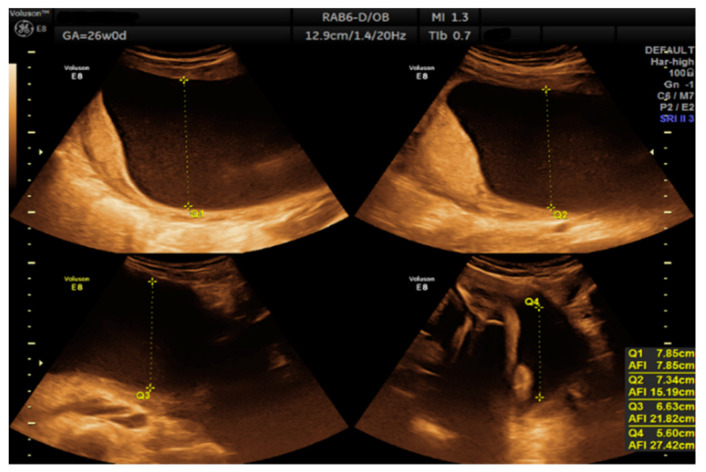
Measurement of the amniotic fluid index at 26 weeks.

**Figure 2 diagnostics-12-01340-f002:**
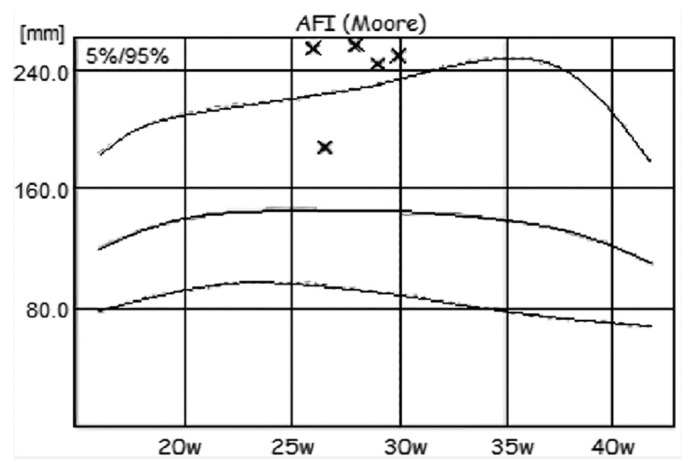
Moore nomogram for AFI values (×).

**Figure 3 diagnostics-12-01340-f003:**
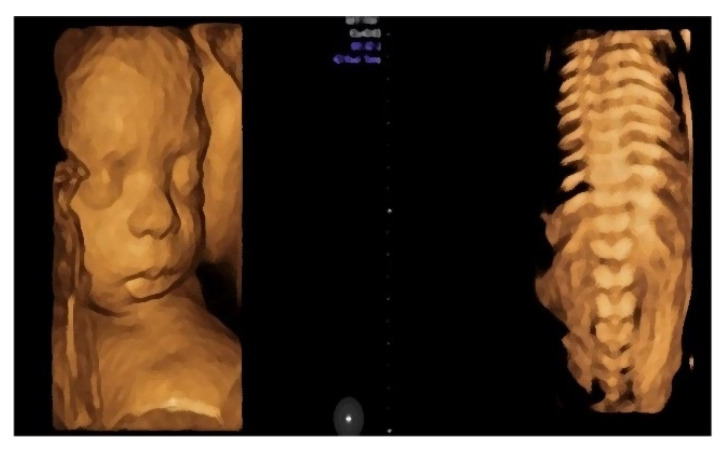
Aspects of the fetus in the second trimester of pregnancy obtained by three-dimensional ultrasonic examination (3D) or so-called anatomical examination.

**Figure 4 diagnostics-12-01340-f004:**
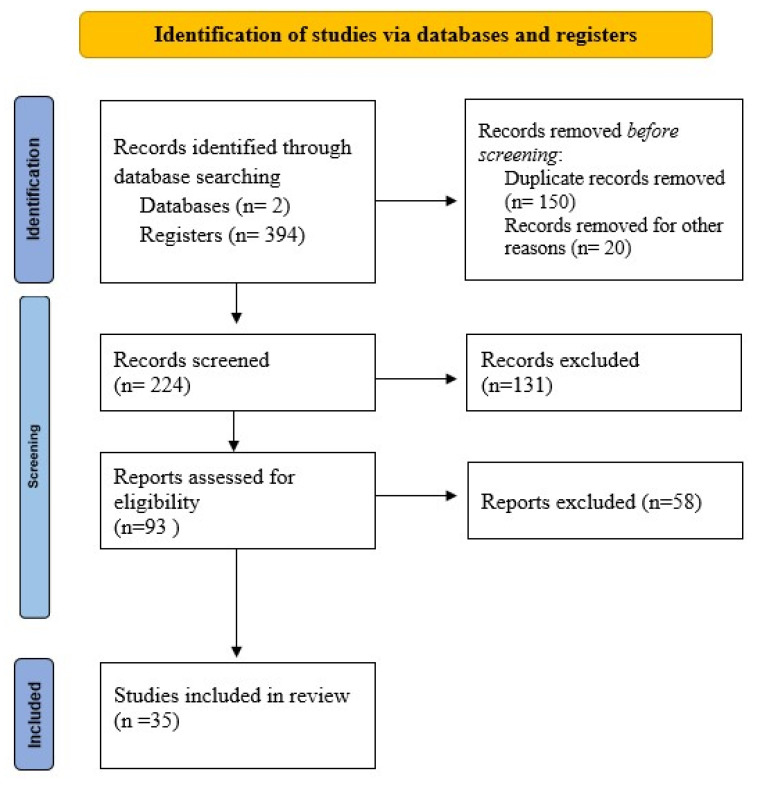
PRISMA 2020 flow diagram for reporting of systematic reviews. Delineation of study selection.

**Table 1 diagnostics-12-01340-t001:** Centralization of data of studies reporting polyhydramnios.

First Author	Number of Patients	Parity	Age (Years)	Polyhydramnios (Max Value)	GA AT Discovery (W)	Cause	GA AT Delivery (W)	Type of Delivery	Birth Weight (G)	APGAR Score (1 min, 5 min)	Maternal Complications	Neonatal Outcome
**Ouyang et al. [14], 2018**	4	N/A	N/A	DVP = 8.1 cm	28	central nervous system abnormalities	N/A	N/A	N/A	N/A	none	Intermittent convulsion.
		N/A	N/A	DVP = 13.2 cm	25	central nervous system abnormalities	N/A	N/A	N/A	N/A	none	Swallowing difficulty; intermittent convulsion, death at 2 months.
		N/A	N/A	DVP = 12.7 cm	32	central nervous system abnormalities	N/A	N/A	N/A	N/A	none	Swallowing difficulty; death at 22 days.
		N/A	N/A	DVP = 13.7 cm	32	central nervous system abnormalities	N/A	N/A	N/A	N/A	none	Breathing difficulty; death at birth.
**Desai et al. [15], 2020**	1	1	26	DVP = 11.5 cm	21	Chorioangioma	32	Emergency C-section	1560	6, 8	none	Respiratory distress with NICU admission for 14 days.
**Iwahata et al. [16], 2021**	1	1	39	AFI = 26 cm	26	old placental chorioangioma	38	Vaginal delivery	2886	9, 10	none	Favorable.
**Singh et al. [17], 2021**	1	1	25	present	24	Chorioangioma	30	Emergency C-section	1200	7, 9	none	Favorable.
**Sakaria et al. [18], 2021**	2	1	21	AFI = 35 cm	22	Kagami-Ogata Syndrome	28	Emergency C-Section	N/A	2, 4	none	Anasarca; micrognathia; large omphalocele containing part of the liver and the bladder; severe respiratory acidosis; life-sustaining measures were discontinued at 2.5 months of life.
		N/A	N/A	N/A	N/A	Kagami-Ogata Syndrome	37	Emergency C-Section	3000	3, 6	none	Flattened nasal bridge; short limbs; cryptorchidism; hypotonia; respiratory distress; coat-hanger appearance of the ribs.
**Thakur et al. [19], 2020**	3	1	23	AFI = 34 cm	28	Bartter syndrome	36	Vaginal delivery	2060	stillborn	none	Stillborn neonate.
**Altmann et al. [20], 2020**	1	1	30	AFI = 31 cm	20	Kagami-Ogata Syndrome	35	Elective C-Section	3660	6, 7	none	Abnormal shape of the thorax; facial dysmorphism; need for ventilation; generalized muscular hypotonia.
**Huang et al. [21], 2019**	1	1	38	present	19	Kagami-Ogata Syndrome	35	Emergency C-Section	3188	3, 6	none	Generalized hypotonia; a flat nasal bridge; respiratory distress; omphalocele; abnormal swallowing function; morphological abnormality of the ribs; facial dysmorphism.
**Choi et al. [22], 2019**	5	N/A	N/A	present	N/A	Costello Syndrome	34	Emergency C-Section	2560	4, 6	none	Atrial tachycardia; feeding problems; growth retardation; cardiac structural anomalies; respiratory distress.
		3	N/A	present	N/A	Costello Syndrome	37	Emergency C-Section	4420	1, 4	none	Atrial tachycardia; feeding problems; cardiac structural anomalies; respiratory distress; growth retardation.
		1	N/A	present	N/A	Costello Syndrome	36	Emergency C-Section	3700	7, 10	none	Atrial tachycardia; feeding problems; cardiac structural anomalies; respiratory distress; growth retardation.
		2	N/A	N/A	N/A	Costello Syndrome	37	Emergency C-Section	4700	N/A	none	Atrial tachycardia; feeding problems; growth retardation; cardiac structural anomalies; respiratory distress; growth retardation.
		N/A	N/A	present	N/A	Costello Syndrome	31	Emergency C-Section	2290	N/A	none	Atrial tachycardia; feeding problems; growth retardation; cardiac structural anomalies; respiratory distress; growth retardation.
**Nam et al. [23], 2021**	1	1	33	AFI = 45 cm	27	Antenatal Bartter syndrome	36	Elective C-Section	2210	9, 10	Aggravating dyspnea; abdominal distension	Hyponatremia; hypokalemia; elevation of plasma renin and aldosterone; 3 months follow-up–good clinical condition.
**Meyer et al. [24], 2018**	1	2	36	AFI = 49 cm	21	Bartter syndrome	29	Vaginal delivery	N/A	N/A	none	Severe polyuria; elevated serum aldosterone and renin activity levels; hydronephrosis.
**Kim et al. [25], 2019**	1	2	37	N/A	27	Infantile cortical hyperostosis	27	Emergency C-section	1970	N/A	none	Severe prematurity; persistent pulmonary hypertension; liver failure; cortical thickening. uncontrolled sepsis on day 38 with death.
**Arthuis et al. [26], 2019**	1	1	23	AFI = 64 cm	19	Antenatal Bartter syndrome	36	Vaginal delivery	3575	N/A	none	Right aortic arch; retro-esophageal left; subclavian artery; moderate pulmonary stenosis; 12 months follow-up–good clinical condition.
**Carvalho et al. [27], 2020**	1	3	37	AFI = 35 cm	3	Isolated Pierre Robin sequence	37	Emergency C-section	2760	5, 7	none	Micrognathia; posterior cleft palate; glossoptosis.
**Yamaguchi et al. [28], 2019**	1	1	31	AFI = 47 cm	24	Myotonic dystrophy	38	Elective C-Section	2838	5, 6	myotonic dystrophy	Distal dominant hypotonia; weakness of breathing; swallowing dysfunction.
**Sakamoto et al. [29], 2019**	1	1	33	AFI = 28 cm	28	duodenal and esophageal atresia without tracheo-esophageal fistula	36	Emergency C-section	2860	8, 10	none	Duodenal and esophageal atresia without tracheo-esophageal fistula; good clinical condition after surgical treatment.
**Wang et al. [30], 2019**	1	3	35	AFI = 30.7 cm	29	24 LMOD3 mutation-positive case of nemaline myopathy	37	Vaginal delivery	N/A	5, 6	none	Stiffness of limbs, little movement; died of respiratory failure 2 days after birth.
**Gica et al. [31], 2019**	1	N/A	29	AFI = 36 cm	25	esophageal atresia	35	N/A	2400	N/A	GDM	Cardiomegaly; atypical esophageal atresia.
**Ardabil et al. [32], 2020**	1	1	34	AFI = 75 cm	33	the midaortic syndrome	33	Emergency C-section	2140	N/A	none	Respiratory distress; biventricular myocardial hypertrophy; renal artery stenosis on the left.
**Willis et al. [33], 2019**	1	3	27	present	30	Chorioangioma	33	Emergency C-section	2850	6, 9	GDM	Fetal anemia; non-immune hydrops fetalis.
**Murata et al. [34], 2020**	1	1	29	AFI = 31 cm	27	Prader-Willi Syndrome	38	Elective C-Section	2492	6, 6	none	Severe hypotonia; cryptorchidism.
**Nabeshima et al. [35], 2020**	1	2	30	AFI = 41 cm	35	22qDS syndrome	38	Emergency C-section	2377	9, 10	none	Small ventricular septal defect; right aortic arch; severe dysphagia.
**Molinet COLL et al. [36], 2020**	1	3	35	AFI = 45 cm	30	Kagami-Ogata Syndrome	36	Emergency C-Section	2050	N/A, 8	none	Respiratory distress; died at 41 days of life.
**Mata et al. [37], 2019**	1	2	36	AFI = 35 cm	33	congenital mesoblastic nephroma	35	Emergency C-Section	2150	8, 9	none	Congenital mesoblastic nephroma.
**Che et al. [38], 2021**	1	1	29	AFI = 25.3 cm	31	congenital mesoblastic nephroma	38	Vaginal delivery	3250	5, N/A	none	Congenital mesoblastic nephroma.
**Takemori et al. [39], 2021**	1	2	35	AFI = 30 cm	29	Bartter syndrome	32	Emergency C-Section	2125	6, 7	none	Serum electrolyte imbalance; polyuria; retinopathy of prematurity.
**Lai et al. [40], 2021**	1	1	34	present	30	VACTERL syndrome	36	Emergency C-section	1832	N/A	none	Esophageal atresia with distal tracheoesophageal fistula; death on day 4 postpartum.
**Katsura et al. [41], 2019**	2	N/A	31	AFI = 30 cm	31	fetal duodenal atresia	36	Emergency C-section	2282	7, 9	none	Fetal duodenal atresia.
		N/A	38	AFI = 31.5 cm	30	fetal duodenal atresia	34	Emergency C-section	2086	1, 3	none	Fetal duodenal atresia.
**Zhang et al. [42], 2018**	2	1	32	AFI = 30.2 cm	28	fetal urinoma	41	Vaginal delivery	3820	7, 9	none	Right renal dysplasia; hydronephrosis; pyelo-ureteric junction obstruction.
		2	33	present	31	fetal urinoma	37	Vaginal delivery	3870	N/A	none	Persistent fetal urinoma.
**Arguello et al. [43], 2021**	1	N/A	N/A	AFI = 36 cm	30	fetal pyloric atresia	33	Emergency C-section	1925	N/A	none	Down syndrome; respiratory distress; fetal type C pyloric atresia.
		1	N/A	AFI = 26.2 cm	25	Bartter syndrome	32	Vaginal delivery	1760	N/A	none	Developmental delay.
		1	25	AFI = 34 cm	25	Bartter syndrome	30	Vaginal delivery	1700	N/A	none	Neonatal death on day 6.
**Baro et al. [44], 2020**	1	2	26	AFI = 37 cm	N/A	GDM	35	Emergency C-section	4030	9, 10	GDM	Sacrococcygeal teratoma.
**Perri et al. [45], 2020**	1	N/A	N/A	present	35	Congenital defects	35	Emergency C-section	2280	2, 5	none	Aortic coarctation; tracheal agenesis; death 150 min postpartum.
**Nunes et al. [46], 2021**	1	1	37	AFI = 25 cm	28	Schaaf-Yang Syndrome	40	Emergency C-section	3000	8, 8	none	Clubfoot; bilateral clindactyly; global hypotonia; distal arthrogryposis.
**Traisrisilp et al. [47], 2021**	1	2	40	AFI = 30 cm	28	Prader-Willi Syndrome	38	Emergency C-section	2420	8, 6	none	Hypotonia; mild chest retraction; difficult feeding.
**Wu et al. [48], 2021**	1	1	27	AFI = 35 cm	21	Bartter syndrome	35	Vaginal delivery	2800	10, 10	none	Favorable.

N/A, not available; GA, gestational age; DVP, deepest vertical pocket; AFI, amniotic fluid index; GDM, gestational diabetes mellitus.

**Table 2 diagnostics-12-01340-t002:** Characteristics of patients.

Age (years old)	31.58 ± 5.04
Gestational age at discovery (weeks)	27.47 ± 4.23
Gestational age at delivery (weeks)	34.93 ± 3.06
Birth weight (g)	2676.9 ± 797.23
Apgar score 1 min	5.96 ± 2.53
Apgar score 5 min	7.5 ± 2.09
AFI (cm)	36.47 ± 11.02
DVP (cm)	12.03 ± 2.06
Parity > 1	40%
Emergency C-Section	64.28%

Values are expressed as *n* (%) or mean ± standard deviation (SD).

**Table 3 diagnostics-12-01340-t003:** The causes of polyhydramnios in the included studies.

Causes of Polyhydramnios	Number of Studies	Expressed as Percentage
**Placental tumors**	4	11.42%
**Gestational diabetes mellitus**	1	2.85%
**Central nervous system abnormalities**	1	2.85%
**Musculoskeletal Disorders (MSDs)**	3	8.57%
**Congenital abnormalities of the kidneys and the urinary tract**	9	25.71%
**Congenital anomalies**		
Gastrointestinal tract	4	11.42%
Others	4	11.42%
**Genetic Disorders**		
Kagami-Ogata syndrome	4	11.42%
Costello syndrome	1	2.85%
Prader-Willi syndrome	2	5.71%
Others	2	5.71%

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
