# Peer review of "Transient Polyhydramnios during Pregnancy Complicated with Gestational Diabetes Mellitus: Case Report and Systematic Review"

_diagnostics, 2022, doi:10.3390/diagnostics12061340_

Round 1

Reviewer 1 Report

In this manuscript authors present a case report of a patient with transient polyhydramnios associated with gestational diabetes mellitus and obstetric complications. Moreover, they performed a systematic review of the literature on polyhydramnios to identify the most common etiological causes and maternal-fetal outcome.

This study is very interesting and only few changes are needed:

  • Figure 2: Quality is very low, resolution must be improved 
  • line 118:  b-HCG must be replaced with  ß-HCG
  • An accurate revision of grammar syntax is recommended

Author Response

Dear Reviewer,

We would like to thank you for your kind review. We have made the following changes as per your suggestions:

Figure 2 – Moore Nomogram has been replaced with a higher quality image.

Line 118 – b-HCG has been replaced to β-HCG

The grammar and spelling of the manuscript have been checked by a native English speaker.

Kind regards,

The authors.

Reviewer 2 Report

1. Make the abstract into one paragraph.
2. Make it clear what is novel or new in this work, even if this is review.  
3. HbA1c may not be adequate to address the issue of DM burden.
Suggest to comment about the availability on recent advances such as 
machine learning (e.g., Communications Biology 3 (1), 1-10)
and sub-stratification subjects (e.g., npj Aging and Mechanisms of Disease 6 (1), 1-12) What is their impact on this work. 

Author Response

Dear Reviewer,

We would like to warmly thank you for your suggestions, which we have followed and made the following corrections:

The abstract has been changed to an unstructured format as recommended.

Regarding your second suggestion, we have added a new paragraph (lines 166-174)

Regarding HbA1c not beeing adequate, we agree and we have specified in lines 89-92 that HbA1c values during pregnancy may not provide the apropiate information about the control of diabetes, and that in these cases, glycemic monitoring is of crucial importance.

            Regarding your last suggestions, we have added new discussions regarding the impact of the mentioned topics in our work (lines 288-302)

Kind regards,

The authors.

This manuscript is a resubmission of an earlier submission. The following is a list of the peer review reports and author responses from that submission.

Round 1

Reviewer 1 Report

the recommendations have been fulfilled

Reviewer 2 Report

The manuscript addresses the issue of transient polyhydramnios. The manuscript is well written and the authors have done a great job of reviewing all reported cases in the literature and describing it in the manuscript-which will especially be of a great interest to the readers